# Surface anticoagulation of mechanical heart valves using electrically induced biomimetic glycocalyx: An in-vitro study to assess hemocompatibility and optimal voltage

**Lokeswara Rao Sajja**[1,2☯*], **Aditya Koppula**[2,3☯*], **Thomas Mathew**[1,2☯], **Anugya Bhatt**[4☯]

**1** Department of Cardiothoracic Surgery, STAR Hospitals, Hyderabad, Telangana, India, **2** Department of Clinical Research, Sajja Heart Foundation, Hyderabad, Telangana, India, **3** Department of Physiology, ESIC Medical College and Hospital, Hyderabad, Telangana, India, **4** Division of Thrombosis research, Biomedical Technology Wing, Sree Chitra Tirunal Institute for Medical Sciences and Technology, Thiruvanthapuram, Kerala, India

☯ These authors contributed equally to this work.
* sajjalr@yahoo.com (LKR); aditya2171985@gmail.com (AK)

## Abstract

Mechanical heart valves (MHVs) remain the most durable option for valve replacement, but they are highly thrombogenic and therefore necessitate life-long anticoagulation. We designed a novel MHV assembly based on "Surface Anticoagulation by Electrically induced Biomimetic Glycocalyx" (SAEBG) that imposes a weak negative potential across the valve's blood-contacting surfaces using an implantable pacemaker as an energy source. The concept is inspired by the native vascular endothelium, where the luminal glycocalyx carries a net negative surface charge that repels platelets and proteins. The present study aims to assess hemocompatibility and thromboresistance in our electrically activated MHV. Bileaflet MHVs (n = 9) were immersed in human platelet-rich plasma or whole blood, of which 5 valves were connected to a programmable pulse generator. Control valves (0 V) (n = 4) were compared to those activated at 0.25 (n = 3), and 0.5 V(n = 2). The control/activated valves were immersed in PRP/blood for 30 min under gentle agitation at 35 ± 2 °C. Hemolysis and blood cell integrity in the supernatant fluid were quantified using standard hematology analyses. Valves immersed in PRP were assessed by scanning electron microscopy (SEM). The cell/deposit free area as assessed by SEM was highest with 0.5 V (~96%), while the corresponding free areas for controls and 0.25 V were ~86% and ~58%, respectively. The platelet counts, platelet and coagulation markers were close to the measurement uncertainty ranges for all the experiments. Hemolysis was < 0.1% for all conditions, and the leucocyte and red-cell counts changed by <2%. Imposing a mild electrical potential (~0.5 V) on an MHV reproduces the anti-thrombotic behaviour of vascular endothelium. The electrical field did not injure

**Data availability statement:** All the data all the fully available without restriction. The raw hematological and coagulation studies for each assessment are shown in Tables 2-5. All the SEM images used in the manuscript have been uploaded as a Supplementary information ZIP file. Information on the US patent granted to valve assembly (including the design details) on which this work is based is available in the public domain (https://patents.google.com/patent/US10736726B2/en?oq=US10736726B2).

**Funding:** The author(s) received no specific funding for this work.

**Competing interests:** NO authors have competing interests.

blood cells or activate coagulation pathways, establishing a safe and effective voltage window for the SAEBG valve to enable further swine model experiments.

## Introduction

Heart valve prostheses were reported to be implanted in more than ~280000 patients worldwide in the year 2009, and about 50% of them were mechanical heart valves (MHV) [1]. Since then, there has been only a marginal increase in the usage of MHV due to a preference for bioprosthetic valves. Nevertheless, MHVs are critically important for younger patients because of their durability and superior mechanical performance [2]. Although they are mechanically durable, their synthetic surfaces trigger platelet adhesion and coagulation, so patients must take long-term anticoagulants such as warfarin [3]. The need for life-long anticoagulation increases the risk of bleeding and interferes with activities of daily living; suboptimal therapy carries a risk of thromboembolism, while over-anticoagulation (higher prothrombin time and international normalized ratio) leads to hemorrhage [4]. Approaches to improve hemocompatibility, such as pyrolytic carbon surfaces or heparin and hydrophilic polymer coatings, reduce but do not abolish thrombogenicity, and coatings may degrade over time [5–7]. A breakthrough would be a durable MHV that resists thrombus formation without systemic anticoagulation.

The vascular endothelium naturally maintains blood fluidity through a carbohydrate-rich glycocalyx on its luminal surface. This layer contains negatively charged glycosaminoglycans, including hyaluronic acid, chondroitin sulphate, and heparan sulphate. The negative surface charge of endothelial cells arises from the glycocalyx and specialised membrane lipids; this electrostatic barrier repels platelets and limits protein adsorption, contributing to the anti-adhesive nature of the endothelium. Several reviews have elaborated on the structure and functions of glycocalyx [8–11]. Artificial surfaces, by contrast, typically become coated with plasma proteins that trigger platelet adhesion and activation [12]. Sawyer and colleagues demonstrated that cathodically polarized vascular grafts resisted thrombosis, suggesting that an imposed electrical field could restore hemocompatibility [13–17]. Building on this concept, we designed an assembly of a bileaflet MHV made of pyrolytic carbon with a pacemaker lead connected to the housing of the valve, which in turn is connected to the pulse generator that delivers a programmable potential to the blood-contacting surfaces.

The present study was conducted to evaluate the hemocompatibility and thrombo-resistance of the electrically activated mechanical heart valve. For a proof of concept, we explore the antithrombotic effect at two specific voltages, i.e., 0.25 and 0.5 V. These voltages were chosen for the following reasons: 1. The normal vascular endothelium in humans is antithrombotic at a potential of 0.3 V due to negatively charged residues in glycocalyx. The choice of 0.25 & 0.5 V is motivated in part by the proximity of these values to the physiological endothelial potential in health. 2. The present work is based on the assembly of a prosthetic mechanical heart valve and the commercially available pacemakers (with leads). The lowest voltage of the commonly used pacemakers is 0.25 V (St. Jude pacemaker, now Abbott) or 0.5 V (Medtronic). 3. Higher

voltage at the valves is likely to increase the risk of myocardial capture (inadvertent action potential genesis in myocardium due to pacemaker activation of the valve) as the threshold pacing voltage is approached, potentially affecting rhythm genesis or propagation. 4. Higher voltage also increases the rate of battery discharge, shortening the duration of antithrombotic efficacy when used in a clinical context, requiring frequent pulse generator replacement. This concept of mechanical prosthetic heart valve assembly was granted a United States of America Patent (US10736726B2) in the year 2020 [18].

## Materials and methods

The in vitro studies were conducted at the Division of Thrombosis Research, Biomedical Technology wing, Sree Chitra Tirunal Institute for Medical Sciences and Technology, Thiruvanthapuram, India, between 14/02/2017 and 29/06/2019.

### Prototype valve assembly

The SAEBG prototype consisted of a standard 25 mm bileaflet mechanical heart valve (*Standard St. Jude Medical mechanical heart valve, St. Paul, Minnesota, USA*) with pyrolytic carbon discs and a titanium housing. A Cobalt-Chromium-Molybdenum (CoCrMo) alloy electrode (*St. Jude Medical Tendril Pacing Lead, Sylmar, California, USA*) was welded to the valve housing through a fenestration made in the sewing ring of valve and the lead was covered with a PTFE tube graft, which was sutured to the margins of the fenestration of the sewing ring, the lead was connected to an external pacemaker-derived generator (*St. Jude Medical ENDURITY CORE, Sylmar, California, USA*). The prosthetic valve-lead-pacemaker circuitry is shown in Fig 1 (A and B). The generator delivered monophasic square pulses at pre-programmed amplitudes (0.25 V and 0.5 V @ 0.4 ms pulse duration) and a frequency of 70 pulses min$^{-1}$. Control heart valves were identical, but not connected to the lead.

### Blood collection and preparation

Fresh human whole blood was obtained from healthy volunteers under institutional approval (First approval: SCT/IEC/594/April-2014, Re-approval: SCT/IEC/1366/April-2019) and anticoagulated with citrate phosphate dextrose. Platelet-rich plasma (PRP) was prepared by centrifugation at 2500 rpm for 5 min; platelet-poor plasma (PPP) was obtained by centrifugation at 4000 rpm for 15 min. Whole blood for hemolysis assays was stored at room temperature and processed within one hour.

### In-vitro exposure and voltage titration

All the in vitro testing procedures conformed to the International Organization for Standardization (ISO) guidelines laid down in the document on Biological evaluation of medical devices – Part 4: Selection of tests for interactions with blood (ISO 10993–4:2017) [19]. The in vitro experiments exposed the test/control valves to blood products for 30 minutes, and the supernatant blood/plasma was tested for cell counts and coagulation markers at two time points. The first (initial) sample is the supernatant fluid collected immediately following the first contact of the valve with blood/plasma, and a second (final) sample was aliquoted 30 minutes after the immersion. The three in vitro experiments were done on different days as follows: 1) Uncharged control valve (V = 0) with PRP, 2) Two test valves (one each at 0.25 and 0.5 V) and one uncharged control valve with PRP, 3) Three test valves (two at 0.25 V and one at 0.5 V) and 2 uncharged control valves with whole blood. Valve discs were placed in 50-ml polystyrene test tubes containing 5 ml phosphate-buffered saline and gently agitated for 5 min to remove surface contaminants. After decanting the saline, 20 ml of PRP (or 40 ml of whole blood for hemolysis assays) was added to the tubes containing the valve discs. Around 5 mL of PRP (or 10 mL of blood) was aspirated immediately for the initial analysis (initial sample). The remaining 15 ml PRP (or 30 ml blood) was agitated at 70 ± 5 rpm with an Environ shaker in an incubator at 35 ± 2 °C for 30 min, after which the PRP (or blood) was aspirated and analyzed (final sample). Empty polystyrene tubes filled with PRP (or blood) were used as a reference, and measurements were made from the reference samples at similar time points, i.e., initial and 30 minutes.

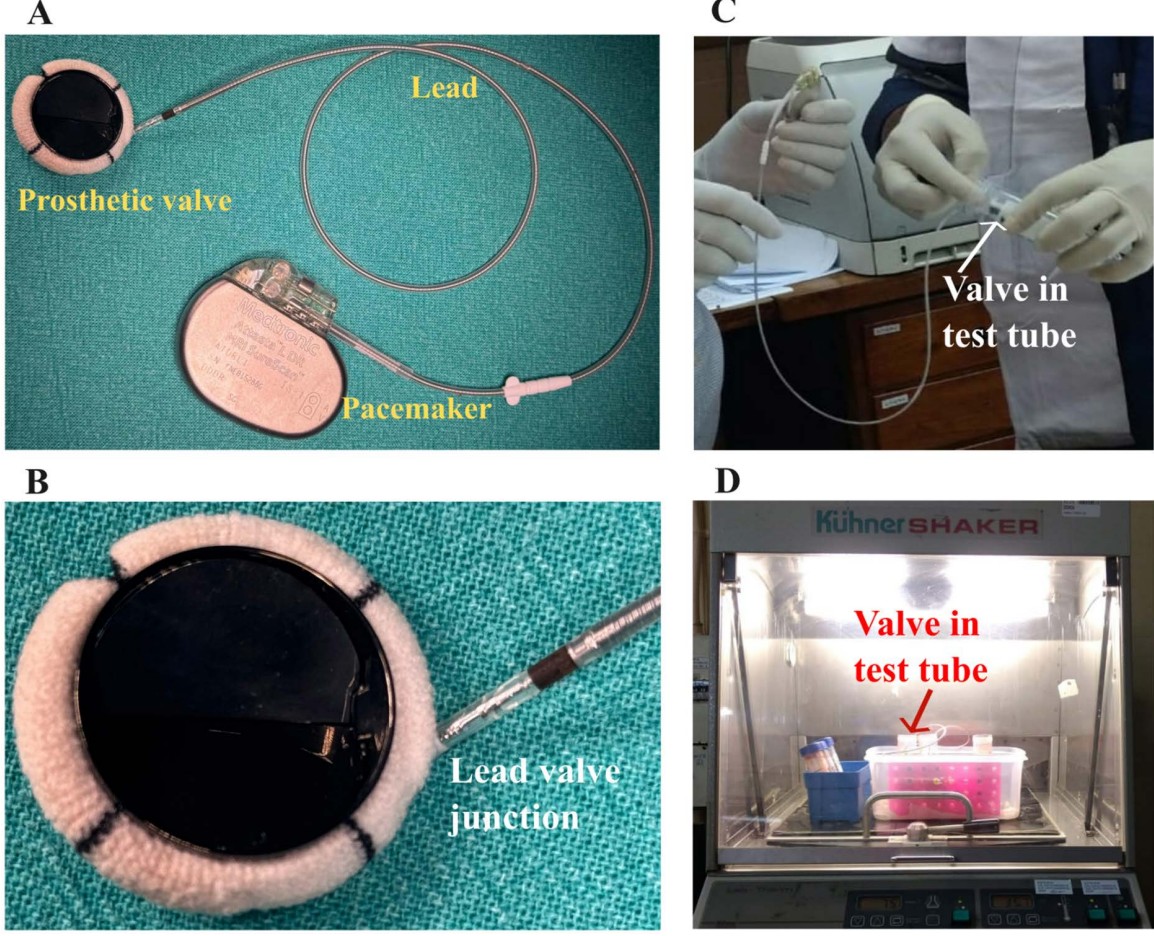

**Fig 1. The experimental setup for testing the antithrombotic effect and hemocompatibility of electrically activated prosthetic valve. A**-Electrical connectivity of the prosthetic valve to the pacemaker with a lead electrode. **B**-Expanded view of prosthetic valve-lead junction. **C**-Prosthetic valve with lead was placed in a 50 ml polystyrene test tube for exposure to blood/PRP, while the pacemaker was placed outside the test tube, in a plastic container (shown in **D**). **D**-Experimental session with blood/PRP exposure to the valve in an environ shaker and an incubator at 35±2 °C for 30 min.**.

## Hematologic and coagulation assays

The initial and final Blood/PRP samples were analyzed for cell counts and coagulation profile before (initial) and after exposure (30 minutes) and the percentage change (initial to 30-minutes) was recorded for test, control, and reference samples. Uncertainty of measurement was reported in percentage (%) units to assess the significance of the experiment-associated changes in the variables.

The Platelet/ Leucocyte/RBC counts, Hematocrit (Hct), and hemoglobin concentration were assessed by Sysmex-K 4500 automated hematology analyzer. The free hemoglobin in the plasma was measured using a Diode array Spectrophotometer. Using the hemoglobin (total Hb), free plasma hemoglobin (free Hb), and Hematocrit (Hct) determined from the methods outlined above, percentage hemolysis was calculated using the following Equation 1 [20]:

$$\% \ Hemolysis = \frac{(100 - Hct\%) * Plasma \ Hb \left(\frac{gm}{dL}\right)}{Total \ Hb \left(\frac{gm}{dL}\right)}$$

$$(1)$$

Platelet activation was assessed using activation markers like P-selectin and platelet factor 4 (PF4) for PRP and blood samples, respectively. For P-selectin, samples were diluted with ACD-PBS to get a final Platelet count of $150-300 \times 10^4$ ml$^{-1}$. Fluorescent-tagged CD26P antibody was added to the sample, incubated for 1 hour and analyzed by flow cytometry using Epics XL, Beckman Coulter. The PF4 was analyzed with commercial ELISA kits (Asserachrom PF4, Diagnostica Stago, France). Absorbance was measured, and ELISA data were processed using the iMark microplate reader and MPM6 software, respectively.

Partial thromboplastin time (PTT) and Fibrinogen levels was measured in the PRP/blood samples using Diagnostica Stago reagent kits on STart 4 Hemostasis Analyzer. Activated complement (C3a) was measured in the samples using commercial ELISA kits (Quidel, USA). Absorbance was measured using BioRad iMark ELISA microplate reader.

### Scanning electron microscopic imaging (SEM)

One PRP-exposed valve for each voltage condition (control, 0.25 V, and 0.5 V) was subjected to SEM imaging. The valves (test and control) were rinsed in buffered saline, fixed in 2% glutaraldehyde, and dehydrated in graded ethanol. The valves were later sputter-coated with gold and subjected to scanning electron microscopy (SEM) at magnifications of up to $5000\times$. Multiple images were acquired for each valve. The SEM images were analyzed using the Fiji distribution of ImageJ software Version 1.54p. Calibration was performed using a linetool and the scale provided on the raw SEM image. Following the calibration, the total visible valve area was measured using a rectangular selection tool on each image, and values were summed over all the images of a particular valve. The RBCs, platelets, and amorphous acellular deposits in each image were manually marked with a polygon tool, fitted with a spline, and the corresponding areas were measured. The absolute areas are reported in $\mu m^2$, and the relative areas are reported as a percentage (%) of the total area. The total area occupied by deposits (cellular/amorphous) and deposit-free areas in each SEM image was computed. The numerical values of areas from all the images in a voltage condition were aggregated and tabulated.

### Data analysis

The raw variable values (initial and 30 minutes) and the percentage change between initial and final values for test, control, and reference samples were computed. Qualitative SEM findings were summarised descriptively. Numerical values of the total valve surface area analyzed, absolute/relative areas of different cellular and amorphous components are tabulated. Because the study objective was to qualitatively characterize and numerically summarize the effect of voltage on a small number of valves, formal statistical comparisons were not performed.

## Results

The number of valves tested for each condition are shown in the Table 1 below:

### Hematologic and coagulation assays

**Platelet counts and activation** Platelet counts changed by less than 10% (Tables 2–5) across all conditions and replicates. Flow-cytometric analysis of P-selectin (CD62P) (Table 3), marker of platelet activation with PRP experiments,

**Table 1. The number of valves tested for each voltage, exposure (blood/PRP) and characterization method (invitro/SEM) (n = 9). One PRP-exposed valve from each voltage condition (0, 0.25 & 0.5 V) was subjected to SEM imaging.**

| S.No | Method | Exposure | Control (0 V) | 0.25 V | 0.5 V |
|------|--------|----------|---------------|--------|-------|
| 1 | Hematologic & coagulation assays | Blood | 2 | 2 | 1 |
| | | PRP | 2 | 1 | 1 |
| 2 | SEM imaging | PRP | 1 | 1 | 1 |

**Table 2. Coagulation profile for the PRP in contact with the Control valve.** The values of various parameters from the aspirated supernatant PRP in contact with the control valve at t = 0 (initial or earliest following contact) and 30 minutes later, along with the % change with valve and the reference sample, are shown. #Platelet response of the reference sample for agonists ADP and collagen was 50% and 61% respectively, on the day of testing. na: measurement uncertainty values for P-selectin are not available.**.

| S.No | Variable | Initial | 30-minutes | % change valve | % change reference | Measurement uncertainty % |
|---|---|---|---|---|---|---|
| 1 | Platelet count (×10⁹ cells/ml) | 3.38 | 3.22 | 4.73 | 1.71 | 10 |
| 2 | P-selectin (% of activated platelets) | 0.02 | 0.02 | 0 | 0 | na |
| 3 | PTT (seconds) | 84 | 89 | 5.95 | 1.43 | 5 |
| 4 | Fibrinogen (g/L) | 2.63 | 2.46 | 6.46 | 1.93 | 5 |
| 5 | C3a (ng/ml) | 880 | 720 | 18.18 | 3.59 | 10 |

**Table 3. The effect of prosthetic valve voltage on the coagulation profile with PRP.** The effect of various prosthetic valve voltages ('0': control valve, 0.25 V and 0.5 V: test valves) on coagulation profiles is compared in PRP-immersed valves. na: measurement uncertainty values for P-selectin are not available.**.

| S.No | Variable | Voltage (V) on the valve | Initial | 30-minutes | % change valve | % change reference | Measurement uncertainty % |
|---|---|---|---|---|---|---|---|
| 1 | P-selectin (% of activated platelets) | 0 | 1.28 | 0.3 | 76.56 | 70.83 | na |
| | | 0.25 | 0.88 | 0.25 | 68.18 | | |
| | | 0.5 | 0.6 | 0.15 | 75 | | |
| 2 | PTT (seconds) | 0 | 149.2 | 150.4 | 0.8 | 3.42 (±1.04) | 5 |
| | | 0.25 | 150.45 | 141.85 | 5.72 | | |
| | | 0.5 | 176.75 | 195.15 | 10.41 | | |
| 3 | Fibrinogen (g/L) | 0 | 2.79 | 2.96 | 6.09 | 3.99 (±0.67) | 5 |
| | | 0.25 | 3.07 | 3.19 | 3.91 | | |
| | | 0.5 | 2.71 | 2.82 | 4.06 | | |
| 4 | C3a (ng/ml) | 0 | 549.42 | 632.89 | 15.19 | 8.17 (±0.68) | 10 |
| | | 0.25 | 305.88 | 396.93 | 29.77 | | |
| | | 0.5 | 479.36 | 560 | 16.82 | | |

showed a similar change in electrically activated valves and the reference sample, indicating no additional risk of platelet activation due to the electrical stimulation of the valves.

**Platelet factor 4 (PF4) secretion** PF4 is a positively charged protein stored in platelet α-granules and is released upon platelet activation. It was used as a marker of platelet activation in blood exposure experiments. At 0.25 V, PF4 concentrations increased from the control valve (0 V) to that at 0.5 V (Table 4), while similar trend was not seen in the replicate experiment (Table 5). The data does not indicate a consistent difference in the platelet activation between the different voltage conditions. Nevertheless, the voltage activation of the valve (0.25 & 0.5 V) does not appear to fully suppress the platelet activation processes indexed by PF4, an implication that is also corroborated by SEM findings (discussed later).

**Partial thromboplastin time (PTT) and fibrinogen** Coagulation parameters remained largely unchanged by the electric field. PTT changes are within or close to the measurement uncertainty for both PRP and blood exposure (Tables 2–5-5), except for the PRP experiment at 0.5 V where the PTT was higher. Similarly, the fibrinogen levels were close to the measurement uncertainty for all the experiments. (Tables 2–5-5).

**Complement activation (C3a)** Complement activation assays did not reveal a consistent trend across the experiments. With whole blood, C3a levels decreased with valves at 0.25 V, 0.5 V, and controls (Tables 4–5). PRP experiments

**Table 4. The effect of prosthetic valve voltage on the coagulation profile with whole blood, Batch 1.** Batch 1 compared the valves at different voltages (0 V-control, 0.25 V, 0.5 V) with the valves immersed in whole blood. Batches 1 and 2 were tested separately on the same day with different reference samples. na: values not available.**.

| S.No | Variable | Voltage (V) on the valve | Initial | 30-minutes | % change valve | % change reference | Measurement uncertainty % |
|---|---|---|---|---|---|---|---|
| 1 | Platelet count (×10⁸ cells/ml) | 0 | 2.75 | 2.53 | 8 | 4.48 (±3.21) | 10 |
| | | 0.25 | 2.69 | 2.63 | 2.23 | | |
| | | 0.5 | 2.96 | 2.84 | 4.05 | | |
| 2 | PF4 (IU/ml) | 0 | 270 | 440 | 62.96 | 8.38 (±1.03) | 10 |
| | | 0.25 | 230 | 430 | 86.96 | | |
| | | 0.5 | 280 | 610 | 117.86 | | |
| 3 | PTT (seconds) | 0 | 206.5 | 200.4 | 2.95 | 3.16 (±1.33) | 5 |
| | | 0.25 | 267 | 261.6 | 2.02 | | |
| | | 0.5 | 233.8 | 233.6 | 0.09 | | |
| 4 | Fibrinogen (g/L) | 0 | 2.6 | 2.65 | 1.92 | 1.91 (±1.85) | 5 |
| | | 0.25 | 2.71 | 2.65 | 2.21 | | |
| | | 0.5 | 2.7 | 2.63 | 2.59 | | |
| 5 | C3a (ng/ml) | 0 | 220.94 | 140.05 | 36.61 | 2.2 (±0.91) | 10 |
| | | 0.25 | 413.8 | 315.3 | 23.8 | | |
| | | 0.5 | 204.99 | 128.69 | 37.22 | | |
| 6 | Leucocyte count (×10⁶ cells/ml) | 0 | 5.8 | 5.9 | 1.72 | 2.3 (±0.95) | 5 |
| | | 0.25 | 5.6 | 5.7 | 1.79 | | |
| | | 0.5 | 5.8 | 6.1 | 5.17 | | |
| 7 | RBC count (×10⁹ cells/ml) | 0 | 4.99 | 5.07 | 1.6 | 0.62 (±0.21) | 5 |
| | | 0.25 | 5.02 | 4.93 | 1.79 | | |
| | | 0.5 | 4.58 | 4.51 | 1.53 | | |
| 8 | Hemoglobin (g/dL) | 0 | 14 | 14.3 | 2.14 | 0.99 (±0.43) | 5 |
| | | 0.25 | 14.1 | 14 | 0.71 | | |
| | | 0.5 | 12.9 | 12.9 | 0 | | |
| 9 | Plasma hemoglobin (mg/dL) | 0 | 6.04 | 11.03 | 82.61589 | 0.77 (±0.35) | na |
| | | 0.25 | 6.08 | 11.48 | 88.81579 | | |
| | | 0.5 | 8.32 | 13.07 | 57.09135 | | |
| 10 | Percentage hemolysis | 0 | na | na | 0.05 | 0.03 (±0.00) | na |
| | | 0.25 | na | na | 0.05 | | |
| | | 0.5 | na | na | 0.06 | | |

showed an increase (Table 3) or a decrease of C3a (Table 2). These conflicting trends could indicate variations in donor complement activity, as opposed to being attributable to the influence of the electrical field.

**Leukocyte counts** Across all voltages and controls, leucocyte counts changed by less than 5% from baseline (Tables 4–5).

**RBC counts, Hemoglobin concentration and hemolysis** Red-cell counts were stable within the measurement uncertainty margin for all the experimental conditions (Tables 4–5). Hemoglobin concentrations (Tables 4–5) remained within ±2% of baseline for all samples with no consistent trend across voltage groups. Plasma haemoglobin (free Hb) increased after exposure (Table 4), possibly due to minor hemolysis, but the large percentage change seen is reflective of low baseline plasma hemoglobin levels. The estimated average percentage hemolysis in all samples was ≤ 0.1%. Typical values ranged from 0.05 to 0.06% at 0.25 & 0.5 V, indicating that electrical activation of the valve is non-hemolytic.

Table 5. The effect of prosthetic valve voltage on the coagulation profile with whole blood, Batch 2. Batch 2 compared the valves at two different voltages (0 V-control and 0.25 V), with the valves immersed in whole blood.**

| S.No | Variable | Voltage (V) on the valve | Initial | 30-minutes | % change valve | % change reference | Measurement uncertainty % |
|---|---|---|---|---|---|---|---|
| 1 | Platelet count (×10$^8$ cells/ml) | 0 | 2.47 | 2.49 | 0.81 | 4.01 (±2.67) | 10 |
| | | 0.25 | 2.91 | 3.02 | 3.78 | | |
| 2 | PF4 (IU/ml) | 0 | 220 | 290 | 31.82 | 5.53 (±2.32) | 10 |
| | | 0.25 | 300 | 370 | 23.34 | | |
| 3 | PTT (seconds) | 0 | 157.8 | 148.3 | 6.02 | 1.31 (±1.09) | 5 |
| | | 0.25 | 215.8 | 222.7 | 3.2 | | |
| 4 | Fibrinogen (g/L) | 0 | 2.92 | 2.8 | 4.11 | 0.94 (±0.4) | 5 |
| | | 0.25 | 2.78 | 2.56 | 7.91 | | |
| 5 | C3a (ng/ml) | 0 | 157.14 | 98.35 | 37.41 | 6.04 (±0.27) | 10 |
| | | 0.25 | 150.31 | 79.97 | 46.8 | | |
| 6 | Leucocyte count (×10$^6$ cells/ml) | 0 | 5.6 | 5.5 | 1.79 | 3.35 (±3.34) | 5 |
| | | 0.25 | 5.5 | 5.7 | 3.64 | | |
| 7 | RBC count (×10$^9$ cells/ml) | 0 | 5.29 | 5.25 | 0.76 | 0.45 (±0.29) | 5 |
| | | 0.25 | 4.51 | 4.57 | 1.33 | | |
| 8 | Hemoglobin (g/dL) | 0 | 14.8 | 14.9 | 0.68 | 0.92 (±0.79) | 5 |
| | | 0.25 | 12.9 | 12.8 | 0.78 | | |
| 9 | Plasma hemoglobin (mg/dL) | 0 | 14.8 | 14.9 | 0.68 | 1.64 (±1.54) | 5 |
| | | 0.25 | 12.9 | 12.8 | 0.78 | | |
| 10 | Percentage hemolysis | 0 | na | na | 0.05 | 0.03 (±0.00) | na |
| | | 0.25 | na | na | 0.06 | | |

From the coagulation and hematologic assays (summarized in Figs 2 and 3), we infer the following: applying a potential at 0.25 V or 0.5 V does not injure blood cells or trigger coagulation. The small changes observed are within the reference variability and are similar to those seen in uncharged controls.

## Morphological surface characterization of valves for Platelets, RBC, and acellular deposits with SEM

SEM images revealed a decrease in adsorption of cells and cell-free amorphous material as the voltage increased from 0.25 to 0.5 V (Figs 4 and 5, Table 6). On valves activated at 0.25 V, amorphous deposits can be seen to cover a substantial portion of the valve surface (Figs 4A & 4B). These deposits are likely adsorbed plasma proteins and/or fibrin-rich clots. Fig 4C shows a large cluster of adherent deformed RBCs. Many cells have collapsed, irregular surfaces, and lack the smooth biconcave contour. Some are shrunken, reflecting membrane damage or dehydration during the processing for SEM. The area of the valve covered by cells/proteinaceous deposits is more than valve at 0.5 V, but paradoxically less than the control valve (Table 6).

At low magnification (Fig 5A), the valve at 0.5 V shows a small number of scattered RBC, but no obvious proteinaceous/fibrin deposits are seen. At higher magnification, some amorphous deposits (Fig 5B) and isolated cells are seen (Figs 5B and 5C). The area covered by cells or protein deposits is the least, with 0.5 V (~4%).

The low magnification SEM image of the control valve (Fig 6A) shows a single cluster of RBCs, and at least three small clusters of platelets at the spreading-dendritic stage indicating activation of platelets [21, 22]. The control valve also shows adherent RBCs and proteinaceous deposits covering ~95% the scanned valve area (Fig 6B and 6C). The SEM images of an unexposed reference valve are shown for comparison in Fig 7.

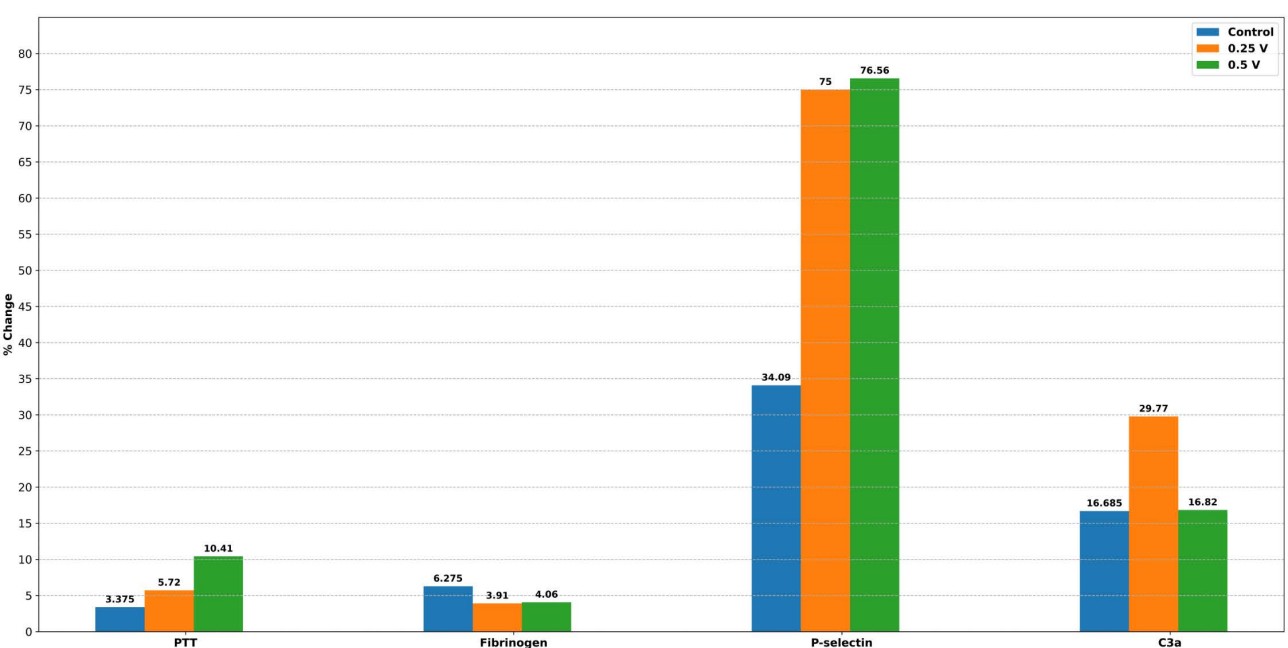

**Fig 2. The effect of prosthetic valve voltage on in vitro testing with PRP.** % change from initial to 30-minutes of immersion of the prosthetic valve is shown in the plot. PTT: Partial thromboplastin time, C3a: Activated complement C3. The height of the bar (numerical value displayed above the bar) for control (0 V) corresponds to the average of the two control valve measurements (Tables 2–3), while it depicts the individual valve values for 0.25 and 0.5 V.**

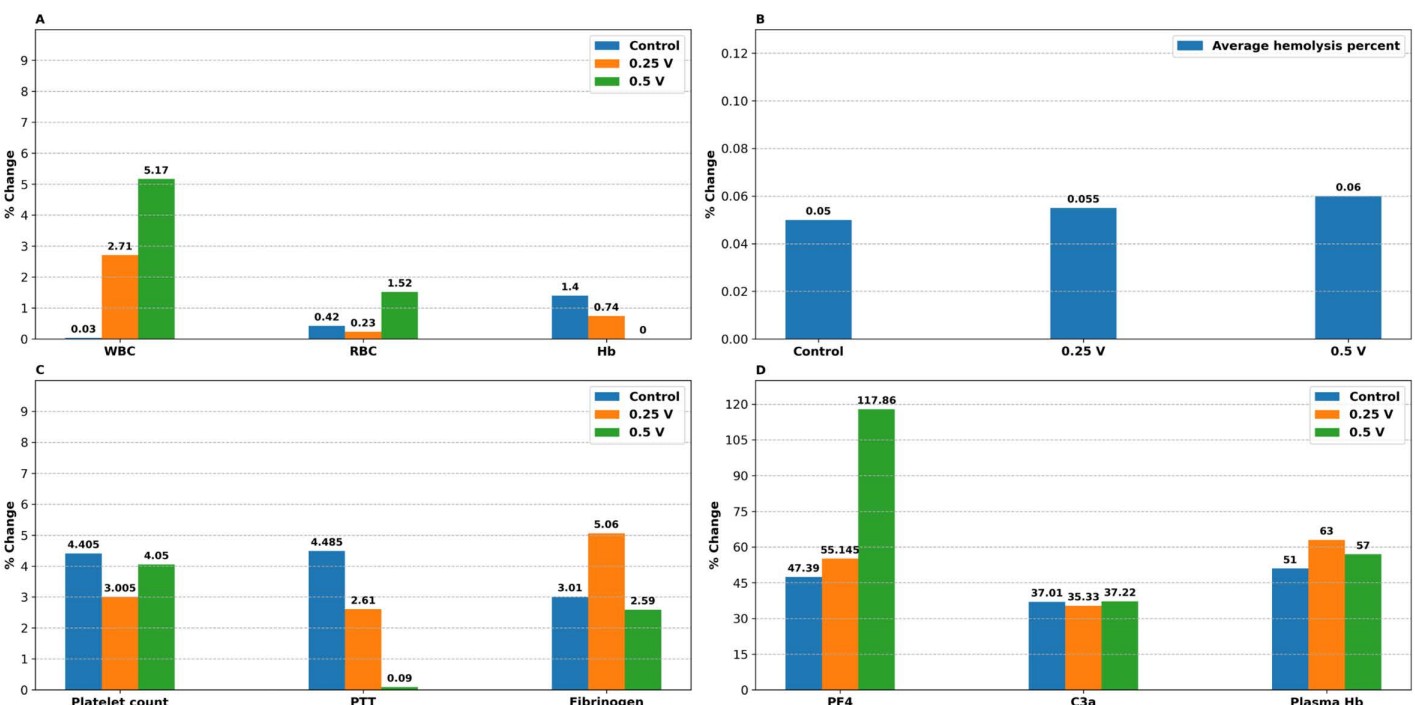

**Fig 3. The effect of prosthetic valve voltage on in vitro testing with Whole blood.** % change from initial to 30 minutes of immersion of the prosthetic valve is shown in the plot. The height of the bars (numerical value displayed above the bar) for control (0 V) and 0.25 V corresponds to the average of two replicate measurements (Tables 4–5), while it depicts the individual values for valve at 0.5 V. PTT: Partial thromboplastin time, C3a: Activated complement C3, Hb: Hemoglobin, PF4: Platelet factor 4. **.

**Fig 4. SEM images of the test valve at 0.25 V and immersed in PRP.** The scanning parameters of the images A, B, C are – **A**: Mag = 2700 ×, WD = 13.1 mm; **B**: Mag = 5000 ×, WD = 13.1 mm; **C**: Mag = 3000 ×, WD = 13.4 mm. Mag = Magnification, WD = working distance.**.

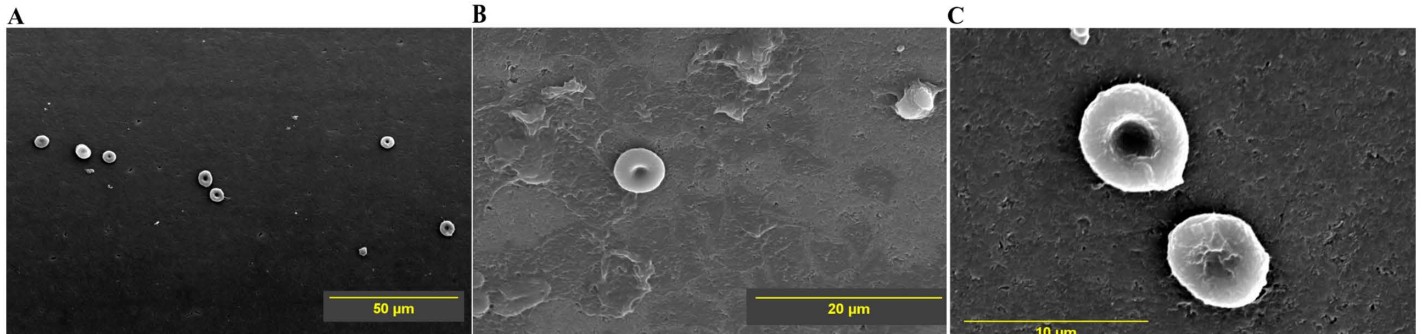

**Fig 5. SEM images of the test valve at 0.5 V and immersed in PRP.** The scanning parameters of the images A, B, C are – **A**: Mag = 1800 ×, WD = 12.4 mm; **B**: Mag = 2500 ×, WD = 12.4 mm; **C**: Mag = 5000 ×, WD = 13.7 mm. Mag = Magnification, WD = working distance.**.

**Table 6. The effect of prosthetic valve voltage on the deposition of cells and acellular proteinaceous coagulum from PRP contact using SEM imaging.** The absolute area (μm²) occupied by RBC, platelets, and acellular deposits is tabulated below. The area numerical values represent the aggregated sum for all the images of a voltage condition. The relative area (%), as a percentage of the total valve area in the images, is shown in parentheses in the same cells. The aggregate relative area (%) of deposits and deposit-free zones is also shown.**.

| S.No | Valve condition | Total valve area assessed (μm²) | RBC area (μm²) | Platelet area (μm²) | Acellular deposits (μm²) | % Area occupied by cells/ deposits | % Cell/ deposit free area |
|---|---|---|---|---|---|---|---|
| 1 | Control (0 V) | 8623.56 | 277.23 (3.22%) | 781.51 (9.06%) | 134.51 (1.56%) | 13.84% | 86.16% |
| 2 | 0.25 V | 7439.90 | 1102.74 (14.82%) | 31.21 (0.42%) | 1962.63 (26.38%) | 41.62% | 58.38% |
| 4 | 0.5 V | 26002.59 | 259.44 (1%) | 21.54 (0.08%) | 768.84 (2.96%) | 4.04% | 95.96% |

We show the measured values of absolute and relative areas of different cellular and amorphous deposits on the valve surface at different voltages in Table 6. Although three images per voltage are shown in the figures (Figs 4-6), the values in Table 6 represent the aggregate of all the images for each voltage condition. All images and

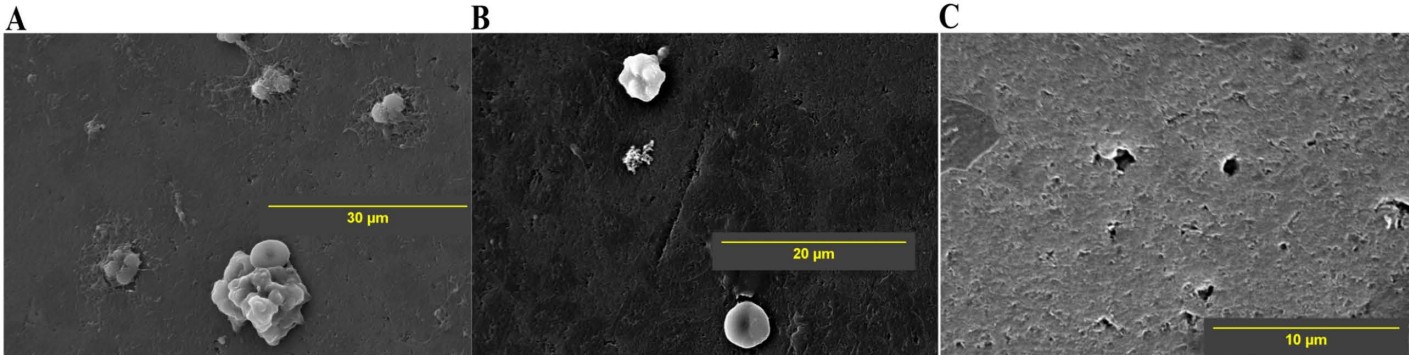

**Fig 6. SEM images of the control valve (0 V) and immersed in PRP.** A, B, C images were taken at multiple sites on the valve with different scanning parameters described as follows – **A**: Mag = 700×, WD = 12.5 mm; **B**: Mag = 2500×, WD = 12.9 mm; **C**: Mag = 5000×, WD = 12.5 mm. Mag = Magnification, WD = working distance**.

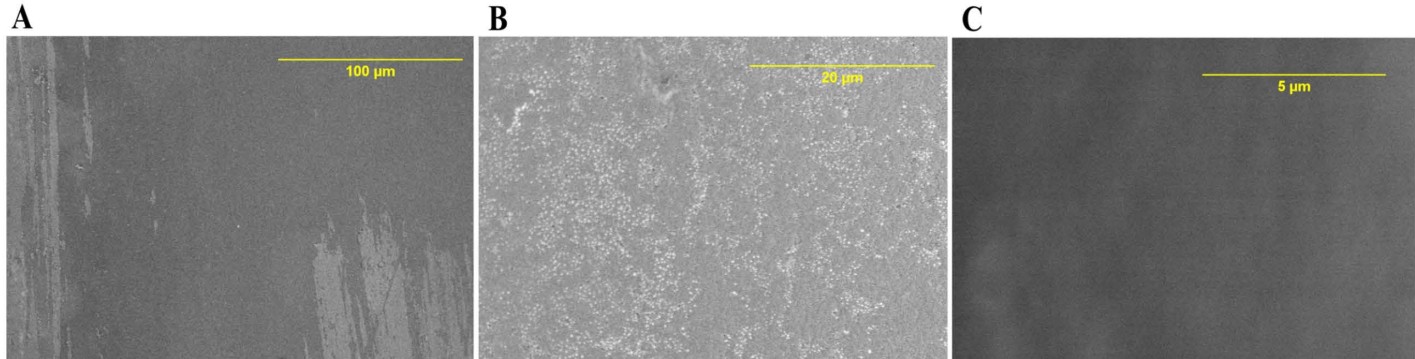

**Fig 7. SEM images of the reference valve that is not exposed to blood/PRP.** A, B, C images were taken at multiple sites on the valve with different scanning parameters described as follows – **A**: Mag = 500×, WD = 10.7 mm, aV = 10 kV; **B**: Mag = 2500×, WD = 10.7 mm, aV = 10 kV; **C**: Mag = 10000×, WD = 9.9 mm, aV = 5 kV. Mag = Magnification, WD = working distance, aV = acceleration voltage, kV = kilovolts. Except for sputter coating artifacts, the valve is free from cellular or proteinaceous deposits, as it is not exposed to blood or PRP**.

their corresponding area measurements are shown in the supplementary information (SEM images at 0.25 V: S1-S10 Figs; 0.5 V: S11-S15 Figs, Control: S16-S25 Figs). It can be seen that the area occupied by cells and/or amorphous deposits decreased as the valve voltage is increased from 0.25 to 0.5 V. In the valves examined, there is a decline in the absolute area occupied by activated platelets with 0.25 and 0.5 V compared to the control valve. The numerical values of absolute/relative areas seem to indicate that the diminished deposition with higher voltage is due to lower adherence of cells (RBC & platelet).

## Discussion

We investigated the hemocompatibility and the potential anti-thrombotic properties of electrically activated prosthetic mechanical heart valves using an invitro approach. We found that the overall cell-adherence, particularly platelets, diminished with the surface voltage at 0.5 V. The deposit-free area increased from 58.38% at 0.25 V to 95.96% at

0.5 V, mostly due to reduced platelet adherence. The voltage-activated valves (0.25 & 0.5 V) were relatively free of activated platelets, while the control (0 V) valve surface had several platelet aggregates in the spreading-dendritic form. The acellular material adsorption was the lowest with the control, highest with 0.25 V, and intermediate with 0.5 V. The Electrical biomimetic glycocalyx (at 0.25 V & 0.5 V) did not eliminate the platelet activation as indicated by the PF4 assay (with whole blood immersion). No consistent differences in coagulation parameters (platelet markers, Partial thromboplastin time, and fibrinogen), cell (platelet, RBC, and leucocyte) counts, and complement activation were found between test and control conditions in the in vitro immersion experiments. Plasma hemoglobin and %hemolysis also did not show a notable difference during the 30-minutes of immersion between the test and valve conditions. These results indicate good hemocompatibility with electrically activated valves, and a minimal tendency to injure blood cells, activate platelets, or trigger coagulation.

Stable platelet counts indicate that the electric field did not cause notable platelet consumption. Lack of consistent difference in platelet and coagulation pathway markers between different voltage conditions implies that the electrical activation of the valve with negative charge does not confer a higher risk of activating platelets and/or coagulation pathways, supporting the safety of this intervention. The shear and agitation-driven platelet degranulation could account for the marginal changes in platelet markers (P-selectin and PF4) [23]. Platelets undergo characteristic morphological transformations as they are progressively activated, viz., *round* (stage I), *dendritic* (stage II), *spreading dendritic* (stage III), *spread* (stage IV), and fully *spread* (stage V) forms [21,22]. The round is the quiescent circulating form, while the fully spread is the completely activated form. The presence of spreading dendritic stage platelet clusters on the control valves, and their virtual absence on the voltage-activated valves, supports the potential effectiveness of the electrical glycocalyx in conferring thromboresistance to the valve and warrants further investigation in larger studies to replicate these results. The monotonous and steep decline in platelet (and RBC) adhesion and activation with the voltage can be explained by greater electrostatic repulsion between the negative zeta potential of the blood cell surface and the more negatively charged valve surface [24]. This is similar to the repulsive electrostatic interaction between the sialic-acid-rich blood cell surface and the endothelial glycocalyx, which is endowed with negatively charged glycosaminoglycans (GAG) [8–11]. The charge-based interaction is known to keep the blood cells away from the endothelium in health. By imposing an electrical potential of ~0.5 V across the valve surface, we reproduced this electrostatic barrier and minimized the tendency for cell adhesion, platelet activation, and cell-based coagulation-driven fibrin deposition [25–27]. In contrast to the effects on cell adhesion/activation, it is also well known that electronegative surfaces increase the contact-based activation of coagulation pathways [28–31]. This could account for intermediate levels of acellular deposition (possibly protein/coagulum) on valves at 0.5 V, reflecting the counteracting effects of diminished cell adhesion, (therefore reduced cell-based coagulation), and increased tendency for direct contact-based coagulation. The trade-off between the contrasting effects of cell adhesion and contact activation at higher surface electronegativity needs to be clarified in future studies.

These findings build on earlier reports that electrically polarised surfaces reduce thrombosis. Sawyer and Pate (1953) [13] made the earliest observations on the role of bioelectric factors in intravascular thrombosis. They studied the effect of electric current applied to electrodes touching the blood vessel wall. The thrombus deposition occurred near the anode (positively charged) but not near the cathode (negatively charged). In subsequent studies [14–17], Sawyer et al. documented a plethora of bioelectric effects viz., a) negative charge on blood vessel wall and blood cells b) reduction and reversal of negative charge after vascular injury at the site of thrombosis c) decrease in negative charge with a fall in pH d) increase in negative charge density with anti-thrombotic drugs and an reversal of polarity with prothrombotic drugs e) enhanced thrombogenecity with positively charged prosthetic materials in contrast to non-thrombogenic nature of negatively charged prosthetic materials f) better nonthrombogenic behaviour with a greater magnitude and more uniform distribution of negative charge. Sawyer's work used cathodic DC [13–16] (direct current

at 200µA)- or high-frequency AC [17] (10µA alternating current at 100 KHz) polarization of metallic grafts, while our study applies a programmable cathodic potential (0.25/0.5 V square pulse train at 70/min) generated by a pacemaker to the pyrolytic carbon mechanical heart valve. This study demonstrates that applying a modest electrical potential to the surfaces of an MHV can emulate the anti-thrombotic properties of the vascular endothelium. Our study indicates that a voltage of 0.5 V ensures thromboresistance and hemocompatibility. The energy requirement for maintaining a 0.5 V potential at 70 bpm is minimal and could potentially offer safe long-term non-pharmacological anticoagulation, free patients from life-long systemic anticoagulant drug usage, improve quality of life, and expand access to durable valve replacement.

Our study has the following limitations. Sample sizes were very small (single valve for some conditions as shown in Table 1), and analysis was descriptive; formal statistical testing was not performed. Findings from the single valve assessment (particularly SEM imaging) need to be verified in larger studies. Hence, our findings are preliminary in nature as the study does not have the statistical power to warrant a generalization of the results, particularly the superiority of 0.5 V over 0.25 V for the antithrombotic effect. It is also possible that there could be individual differences in the optimal voltage (0.25 vs. 0.5 V) for antithrombosis. Nevertheless, a greater anti-thrombotic effect with a higher voltage is a physiologically plausible conjecture that could be assessed in future studies. The optimal valve surface voltage that achieves maximum antithrombosis, minimum or no risk myocardial electrophysiological capture, while having a long battery life (for sustained effect) is still an open question. Differences in the manufacturing characteristics of the valves (possible surface irregularities) could potentially explain the results obtained with respect to the effect of different voltages. However, the valves used in the study are approved and widely used for valve replacement in humans. The occurrence of thrombosis after implantation due to surface defects has not been reported to our knowledge. Furthermore, SEM image assessment did not reveal any gross surface defect that is likely to induce thrombosis in any of the fields examined, making the possibility of a manufacturing defect explaining the study findings very unlikely.

This ex vivo study used human PRP and whole blood under static agitation; the complex hemodynamics of the cardiac outflow tract may influence platelet interactions differently in vivo. Biomarkers such as PF4 and complement C3a are surrogate indicators; the definitive endpoint remains prevention of clinical valve thrombosis. Only a single disc geometry and two voltages above the threshold were tested; other materials or pulse waveforms may alter the optimal potential. The in vitro paradigm precludes the assessment of the electrophysiological effects of voltage applied to the valve on the heart. However, the voltages tested in the study (0.25 & 0.5 V) are unlikely to significantly impact the rhythm genesis or propagation in the cardiac muscle for the following reasons: 1. The threshold voltage for myocardial capture for pacing applications is ~1.5 V @ 0.4 ms pulse width, while we have used the pacemaker in a non-pacing mode at 0.25 & 0.5 V for electrical valve activation. The normal vascular endothelium that lines the inside of the heart/blood vessels has a potential difference of ~0.3 V 2. The valve prosthesis will be implanted in the mitral position, which is surrounded by an electrically insulating annulus fibrosus (the physiological fibrous skeleton of the heart). The annulus fibrosis prevents the direct propagation of electrical activity from atria to ventricles in health, and similar behaviour is expected to protect the heart from the electrophysiological spillover effects. 3. The valve prosthesis has a cuff of sewing ring, made up of polyester fabric, which helps suture the valve to the annulus. This fabric material has electrical insulator properties. 4. Peri-prosthetic valve fibrosis, known to occur after valve implantation, further walls off the prosthetic valve electrically from the rest of the heart. A schematic image indicating the potential method of deployment in the clinical context for which the valve is proposed to be used is shown in Fig 8, including pacemaker connectivity.

Further work will focus on in vivo validation and long-term safety. Swine animal models will be used to assess valve patency over extended periods without systemic anticoagulation. Device miniaturization and integration into implantable pacemakers will be explored to create a fully implantable self-anticoagulant mechanical heart valve.

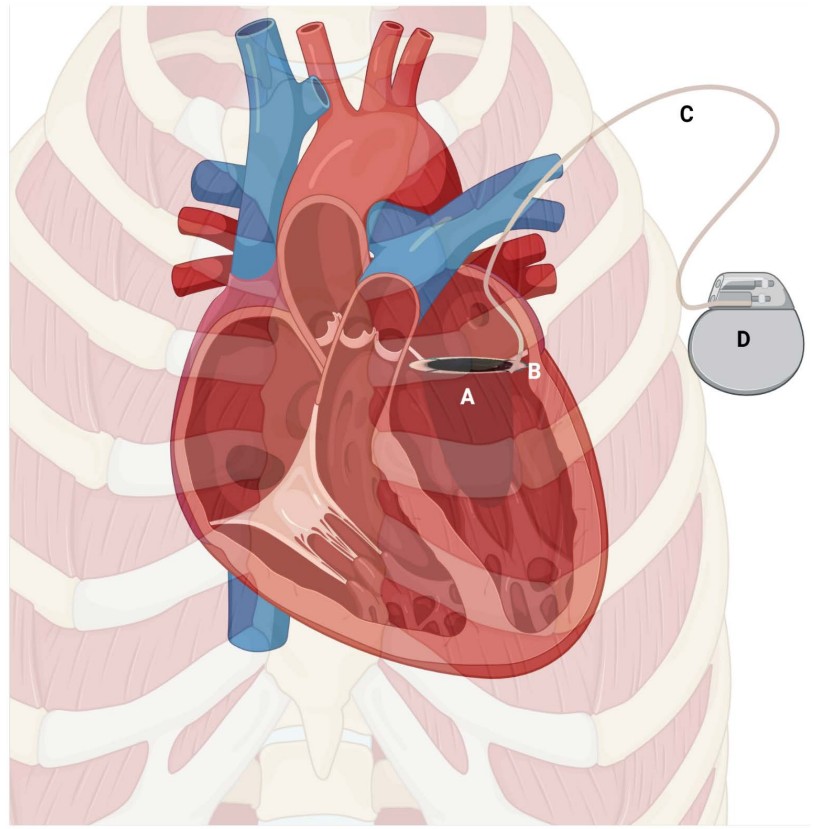

**Fig 8. A schematic representation of the intended clinical deployment of the electrically activated prosthetic valve, including the pacemaker-electrical connectivity. A**- Prosthetic mechanical heart valve. **B**- Valve-lead junction. **C**-Lead electrode. **D**-Pacemaker.

## Supporting information

**S1 Fig. Cells and other deposits are annotated by yellow colored outlines.** Total area = 112.749 µm², platelet area = 31.212 µm² (single platelet in dendritic spreading stage of platelet activation), RBC area = 0, Acellular deposit area = 0, Total area occupied by masses (cells and deposits)=31.212 µm² (27.68%), Mass free area = 81.537 µm² (72.32%), Magnification = 10000x, working distance = 14 mm. The scale bar is shown in the image for reference. (TIFF)

**S2 Fig. Cells and other deposits are annotated by yellow colored outlines.** Mass of shrunken RBCs are seen along with amorphous material deposits. Total area = 444.587 µm², platelet area = 0, RBC area = 62.06 µm², Acellular deposit area = 21.662 µm², Total area occupied by masses (cells and deposits)=83.722 µm² (18.83%), Mass free area = 360.865 µm² (81.17%), Magnification = 5000x, working distance = 13.1 mm. The scale bar is shown in the image for reference. (TIFF)

**S3 Fig. Cells and other deposits are annotated by yellow colored outlines.** Single deformed RBC is seen along with amorphous material deposits. Total area = 452.001 µm², platelet area = 0, RBC area = 21.61 µm², Acellular deposit area = 25.811 µm², Total area occupied by masses (cells and deposits)=47.421 µm² (10.49%), Mass free area = 404.58 µm² (89.51%), Magnification = 5000x, working distance = 13.2 mm. The scale bar is shown in the image for reference. (JPG)

**S4 Fig. Cells and other deposits are annotated by yellow colored outlines.** Single RBC is seen along with amorphous material deposits. Total area = 450.614 µm², platelet area = 0, RBC area = 20.986 µm², Acellular deposit area = 28.163 µm², Total area occupied by masses (cells and deposits)=49.149 µm² (10.91%), Mass free area = 401.465 µm² (89.09%), Magnification = 5000x, working distance = 12.6 mm. The scale bar is shown in the image for reference.
(TIFF)

**S5 Fig. Cells and other deposits are annotated by yellow colored outlines.** A single RBC, along with large areas of amorphous deposits covering the valve, is seen. Total area = 451.497 µm², platelet area = 0, RBC area = 19.736 µm², Acellular deposit area = 298.594 µm², Total area occupied by masses (cells and deposits)=318.33 µm² (70.51%), Mass free area = 133.167 µm² (29.49%), Magnification = 5000x, working distance = 13.1 mm. The scale bar is shown in the image for reference.
(TIFF)

**S6 Fig. Cells and other deposits are annotated by yellow colored outlines.** Large areas of amorphous deposits resembling fibrin coagulum covering the valve are seen. Total area = 1553.502 µm², platelet area = 0, RBC area = 0 µm², Acellular deposit area = 1138.716 µm², Total area occupied by masses (cells and deposits)=1138.716 µm² (73.3%), Mass free area = 414.786 µm² (26.7%), Magnification = 2700x, working distance = 13.1 mm. The scale bar is shown in the image for reference.
(TIFF)

**S7 Fig. Cells and other deposits are annotated by yellow colored outlines.** Large numbers of RBCs forming aggregates are seen covering the valve. Total area = 1267.215 µm², platelet area = 0, RBC area = 570.823 µm², Acellular deposit area = 5.531 µm², Total area occupied by masses (cells and deposits)=576.354 µm² (45.48%), Mass free area = 690.861 µm² (54.52%), Magnification = 3000x, working distance = 13.4 mm. The scale bar is shown in the image for reference.
(TIFF)

**S8 Fig. Cells and other deposits are annotated by yellow colored outlines.** Isolated RBCs and small focal deposits are seen on the valve. Total area = 451.65 µm², platelet area = 0, RBC area = 56.026 µm², Acellular deposit area = 98.267 µm², Total area occupied by masses (cells and deposits)=154.293 µm² (34.16%), Mass free area = 297.357 µm² (65.84%), Magnification = 5000x, working distance = 12.9 mm. The scale bar is shown in the image for reference.
(TIFF)

**S9 Fig. Cells and other deposits are annotated by yellow colored outlines.** Large amorphous material seen on the valve. Total area = 451.499 µm², platelet area = 0, RBC area = 0 µm², Acellular deposit area = 270.202 µm², Total area occupied by masses (cells and deposits)=270.202 µm² (59.85%), Mass free area = 181.297 µm² (40.15%), Magnification = 5000x, working distance = 12.8 mm. The scale bar is shown in the image for reference.
(TIFF)

**S10 Fig. Cells and other deposits are annotated by yellow colored outlines.** Clusters of RBCs and small focal deposits are seen on the valve. Total area = 1804.586 µm², platelet area = 0, RBC area = 351.496 µm², Acellular deposit area = 75.685 µm², Total area occupied by masses (cells and deposits)=427.181 µm² (23.67%), Mass free area = 1377.405 µm² (76.33%), Magnification = 2500x, working distance = 12.3 mm. The scale bar is shown in the image for reference.
(TIFF)

**S11 Fig. Cells and other deposits are annotated by yellow colored outlines.** Isolated RBC, smaller platelet aggregate, and amorphous deposits are seen on the valve. Total area = 1800.529 µm², platelet area = 21.542, RBC area = 21.64 µm², Acellular deposit area = 529.43 µm², Total area occupied by masses (cells and deposits)=573.393 µm² (31.85%),

Mass free area = 1227.136 µm² (68.15%), Magnification = 2500x, working distance = 12.9 mm. The scale bar is shown in the image for reference.
(TIFF)

**S12 Fig. Cells and other deposits are annotated by yellow colored outlines.** Amorphous deposits resembling RBC ghosts were seen on the valve. Total area = 452.173 µm², platelet area = 0, RBC area = 0, Acellular deposit area = 219.492 µm², Total area occupied by masses (cells and deposits)=219.492 µm² (48.54%), Mass free area = 232.681 µm² (51.46%), Magnification = 5000x, working distance = 12.9 mm. The scale bar is shown in the image for reference.
(TIFF)

**S13 Fig. Cells and other deposits are annotated by yellow colored outlines.** A single RBC with finger like cytoplasmic projections, presumably due to dehydration, is seen. Total area = 265.16 µm², platelet area = 0, RBC area = 16.573 µm², Acellular deposit area = 0, Total area occupied by masses (cells and deposits)=16.573 µm² (6.25%), Mass free area = 248.587 µm² (93.75%), Magnification = 6500x, working distance = 13 mm. The scale bar is shown in the image for reference.
(TIFF)

**S14 Fig. Cells and other deposits are annotated by yellow colored outlines.** Two isolated RBCs with wrinkled membranes are seen. Total area = 450.215 µm², platelet area = 0, RBC area = 51.721 µm², Acellular deposit area = 0.807, Total area occupied by masses (cells and deposits)=52.528 µm² (11.67%), Mass free area = 397.687 µm² (88.33%), Magnification = 5000x, working distance = 13 mm. The scale bar is shown in the image for reference.
(JPG)

**S15 Fig. Cells and other deposits are annotated by yellow colored outlines.** Few solitary RBCs and small deposits are seen, but much of the valve area is virtually deposit-free. Total area = 23034.515 µm², platelet area = 0, RBC area = 169.504 µm², Acellular deposit area = 18.334, Total area occupied by masses (cells and deposits)=187.838 µm² (0.82%), Mass free area = 22846.68 µm² (99.18%), Magnification = 700x, working distance = 12.5 mm. The scale bar is shown in the image for reference.
(TIFF)

**S16 Fig. Cells and other deposits are annotated by yellow colored outlines.** A solitary RBC and a cluster of RBCs are seen. One or more platelets in the spreading dendritic stage are seen. Total area = 561.36 µm², platelet area = 20.105, RBC area = 68.779 µm², Acellular deposit area = 0, Total area occupied by masses (cells and deposits)=88.88 µm² (15.83%), Mass free area = 472.476 µm² (84.17%), Magnification = 4500x, working distance = 12.7 mm. The scale bar is shown in the image for reference.
(TIFF)

**S17 Fig. Cells and other deposits are annotated by yellow colored outlines.** A cluster of activated platelets in the spreading dendritic stage is seen in the centre of the field. Total area = 447.435 µm², platelet area = 44.362, RBC area = 0 µm², Acellular deposit area = 19.548, Total area occupied by masses (cells and deposits)=63.91 µm² (14.28%), Mass free area = 383.525 µm² (85.72%), Magnification = 5000x, working distance = 12.6 mm. The scale bar is shown in the image for reference.
(TIFF)

**S18 Fig. Cells and other deposits are annotated by yellow colored outlines.** A solitary RBC is seen in the center of the field. Total area = 447.921 µm², platelet area = 0, RBC area = 18.494 µm², Acellular deposit area = 0, Total area occupied by masses (cells and deposits)=18.494 µm² (4.13%), Mass free area = 429.427 µm² (95.87%), Magnification = 5000x, working distance = 12.6 mm. The scale bar is shown in the image for reference.
(TIFF)

**S19 Fig. Cells and other deposits are annotated by yellow colored outlines.** A large cluster of deformed RBCs is seen in the lower half of the field. At least three clusters of adherent and activated platelets in the spreading dendritic stage are seen in addition to focal amorphous deposits. Total area=3491 µm$^2$, platelet area=508.738 µm$^2$, RBC area=149.973 µm$^2$, Acellular deposit area=85.18 µm$^2$, Total area occupied by masses (cells and deposits)=743.891 µm$^2$ (21.31%), Mass free area=2747.109 µm$^2$ (78.69%), Magnification=1800x, working distance=12.4 mm. The scale bar is shown in the image for reference. (TIFF)

**S20 Fig. Cells and other deposits are annotated by yellow colored outlines.** A larger activated platelet-rich deposit is seen in the field. Total area=449.697 µm$^2$, platelet area=205.294 µm$^2$, RBC area=0, Acellular deposit area=7.403 µm$^2$, Total area occupied by masses (cells and deposits)=213.327 µm$^2$ (47.44%), Mass free area=236.37 µm$^2$ (52.56%), Magnification=5000x, working distance=12.5 mm. The scale bar is shown in the image for reference. (TIFF)

**S21 Fig. Cells and other deposits are annotated by yellow colored outlines.** Two isolated swollen RBC are seen with focal amorphous deposits. Total area=1793.741 µm$^2$, platelet area=0 µm$^2$, RBC area=39.984 µm$^2$, Acellular deposit area=7.977 µm$^2$, Total area occupied by masses (cells and deposits)=47.961 µm$^2$ (2.67%), Mass free area=1745.78 µm$^2$ (97.33%), Magnification=2500, working distance=12.4 mm. The scale bar is shown in the image for reference. (TIFF)

**S22 Fig. Cells and other deposits are annotated by yellow colored outlines.** Total area=448.409 µm$^2$, platelet area=0 µm$^2$, RBC area=0 µm$^2$, Acellular deposit area=0 µm$^2$, Total area occupied by masses (cells and deposits)=0 (0%), Mass free area=448.409 µm$^2$ (100%), Magnification=5000x, working distance=13.7 mm. The scale bar is shown in the image for reference. (TIFF)

**S23 Fig. Cells and other deposits are annotated by yellow colored outlines.** A single platelet extending its dendritic process, indicating the incipient stage of platelet activation. Total area=454.589 µm$^2$, platelet area=2.379 µm$^2$, RBC area=0, Acellular deposit area=0, Total area occupied by masses (cells and deposits)=2.379 µm$^2$ (0.52%), Mass free area=452.21 µm$^2$ (99.48%), Magnification=5000x, working distance=13.7 mm. The scale bar is shown in the image for reference. (TIFF)

**S24 Fig. Cells and other deposits are annotated by yellow colored outlines.** Focal amorphous deposits are seen. Total area=451.483 µm$^2$, platelet area=0, RBC area=0, Acellular deposit area=14.404 µm$^2$, Total area occupied by masses (cells and deposits)=14.404 µm$^2$ (3.19%), Mass free area=437.079 µm$^2$ (96.81%), Magnification=5000x, working distance=13.8 mm. The scale bar is shown in the image for reference. (TIFF)

**S25 Fig. Cells and other deposits are annotated by yellow colored outlines.** Total area=77.924 µm$^2$, platelet area=0, RBC area=0, Acellular deposit area=14.404 µm$^2$, Total area occupied by masses (cells and deposits)=0 (0%), Mass free area=77.924 µm$^2$ (100%), Magnification=12000x, working distance=13.8 mm. The scale bar is shown in the image for reference. (TIFF)

## Acknowledgments

We thank the volunteers who donated blood for this study and the staff of the Biomedical Technology Wing, Sree Chitra Tirunal Institute for Medical Sciences and Technology, Trivandrum, for performing the hemocompatibility assays. This work was supported by the Sajja Heart Foundation.

## Author contributions

**Conceptualization:** Lokeswara Rao Sajja.

**Data curation:** Lokeswara Rao Sajja, Aditya Koppula.

**Formal analysis:** Lokeswara Rao Sajja, Aditya Koppula.

**Funding acquisition:** Lokeswara Rao Sajja.

**Investigation:** Lokeswara Rao Sajja, Thomas Mathew, Anugya Bhatt.

**Methodology:** Lokeswara Rao Sajja, Thomas Mathew, Anugya Bhatt.

**Project administration:** Lokeswara Rao Sajja.

**Resources:** Lokeswara Rao Sajja, Thomas Mathew, Anugya Bhatt.

**Software:** Aditya Koppula.

**Supervision:** Lokeswara Rao Sajja, Thomas Mathew.

**Validation:** Aditya Koppula, Anugya Bhatt.

**Visualization:** Aditya Koppula, Anugya Bhatt.

**Writing – original draft:** Lokeswara Rao Sajja, Aditya Koppula, Anugya Bhatt.

**Writing – review & editing:** Lokeswara Rao Sajja, Aditya Koppula, Thomas Mathew, Anugya Bhatt.

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
