## [Decision Letter · Decision Letter 0]

21 Jan 2026

PONE-D-25-57739Surface Anticoagulation of Mechanical Heart Valves using Electrically Induced Biomimetic Glycocalyx: an In-vitro study to assess Hemocompatibility and Optimal VoltagePLOS One

Dear Dr. Sajja,

Thank you for submitting your manuscript to PLOS ONE. After careful consideration, we feel that it has merit but does not fully meet PLOS ONE’s publication criteria as it currently stands. Therefore, we invite you to submit a revised version of the manuscript that addresses the points raised during the review process.

**ACADEMIC EDITOR:**The Manuscript addresses the major challenge in the field and novel. With additional revision, it can be considered further. We look forward to receive the revised version. 

We look forward to receiving your revised manuscript.

Kind regards,

Jeevithan Elango, PhD

Academic Editor

PLOS One

Journal Requirements:

Additional Editor Comments (if provided):

Find the reviewers' comments and look forward to receive your revised MS for further consideration.

Reviewers' comments:

Reviewer's Responses to Questions

**Comments to the Author**

1. Is the manuscript technically sound, and do the data support the conclusions?

Reviewer #1: Partly

Reviewer #2: Yes

2. Has the statistical analysis been performed appropriately and rigorously? 

Reviewer #1: I Don't Know

Reviewer #2: Yes

3. Have the authors made all data underlying the findings in their manuscript fully available?

Reviewer #1: No

Reviewer #2: Yes

4. Is the manuscript presented in an intelligible fashion and written in standard English?

Reviewer #1: Yes

Reviewer #2: Yes

5. Review Comments to the Author

Reviewer #1: This manuscript developed a unique method, the application of voltage, to make the valve antithrombogenic for valve leaflet. This was a very challenging and interesting topic. However, there are several unclear points in the manuscript that need improvement.

1. Sample photos should be included.

2. To deepen the reader's understanding, it would be better to present a scheme of how the valve is intended to be used in the future including the electrical circuit system.

3. Macroscopic photographs taken after the blood compatibility test should be added to clarify the surface condition.

4. The SEM photographs show enlarged sections. Please also show the overall picture.

5. In the section on limitation, please consider how you think the electrophysiological effects on the heart would be affected if voltage were applied to the valve.

Reviewer #2: An intriguing and extremely pertinent use of bio-electric engineering is examined in the manuscript "Surface Anticoagulation of Mechanical Heart Valves..." It is a clever idea to reject platelets by imitating the endothelium glycocalyx with a mild electrical current. The reasoning is supported by prior research on surface charge and thrombosis, and the prose is usually understandable. As a reviewer, however, I have serious doubts about how reliable the information is. Formal statistical analysis is not possible due to the study's reliance on a relatively small number of replicates (often one valve per condition). As a result, it is challenging to ascertain whether the reported differences—like the apparent superiority of 0.5 V over 0.25 V—are merely the result of random variation or are biologically significant. To meet the strict requirements of the area, this effort must either strengthen the statistical power or, on the other hand, be far more cautious about the assertions.

1-The abstract mentions comparing control valves to 0.25 V and 0.5 V but does not specify the sample size (n)

2-The introduction would benefit from a brief explanation of why the particular voltages of 0.25 V and 0.5 V were selected, even though the background information about the glycocalyx is great. Are these based on safety thresholds for electrochemical corrosion, earlier computer models, or earlier research on vascular grafts (Sawyer et al.)?

3-This is the critical weak point of the manuscript. Section 2.6 states, "formal statistical comparisons were not performed." In a comparative efficacy study, this is a significant deviation from standard practice

4-The valves were exposed to blood/PRP for only 30 minutes. While platelet adhesion occurs early, meaningful thrombus formation and fibrin network development on MHVs typically occur over longer periods.

5-Hemolysis was found to be less than 0.1%, which is a favorable safety signal. But was the donor blood's hematocrit taken into account while calculating the free hemoglobin?

6-According to the manuscript, 0.25 V was "insufficient" and 0.5 V worked well. This is a dangerous conclusion based on the SEM results from a single valve. It's possible that the particular valve used for 0.25 V had a surface flaw or that 0.25 V works in a different donor. These alternate explanations should be thoroughly examined throughout the conversation.

6. PLOS authors have the option to publish the peer review history of their article (what does this mean?). If published, this will include your full peer review and any attached files.

Reviewer #1: **Yes:**Atsushi Mahara

Reviewer #2: **Yes:**Mohammad EL-Nablaway

You may also use PLOS’s free figure tool, NAAS, to help you prepare publication quality figures: https://journals.plos.org/plosone/s/figures#loc-tools-for-figure-preparation

---

## [Author Response · Author response to Decision Letter 1]

16 Feb 2026

Editor comments

1. Please ensure that your manuscript meets PLOS ONE's style requirements, including those for file naming. The PLOS ONE style templates can be found at…

A: We have reviewed and revised the manuscript to meet PLOS ONE’s style requirements, including the specific recommendations for the file naming.

We have modified the abstract as per the guidelines into an unstructured format

We have altered the section heading as per the guidelines

Figure and Table naming for main manuscript and supplementary information have been made compliant with the recommended style of the PLOSone

2. We note that you have indicated that there are restrictions to data sharing for this study. For studies involving human research participant data or other sensitive data, we encourage authors to share de-identified or anonymized data. However, when data cannot be publicly shared for ethical reasons, we allow authors to make their data sets available upon request. For information on unacceptable data access restrictions, please see (http://journals.plos.org/ plosone/s/data-availability#) loc-unacceptable-data-access-restrictions.

a) If there are ethical or legal restrictions on sharing a de-identified data set, please explain them in detail(e.g., data contain potentially identifying or sensitive patient information, data are owned by a third-party organization, etc.) and who has imposed them (e.g., a Research Ethics Committee or Institutional Review Board, etc.). Please also provide contact information for a data access committee, ethics committee, or other institutional body to which data requests may be sent. b) If there are no restrictions, please upload the minimal anonymized data set necessary to replicate your study findings to a stable, public repository and provide us with the relevant URLs, DOIs, or accession numbers. Please see http://www.bmj.com/content/340/bmj.c181.long. for guidelines on how to de-identifyand prepare clinical data for publication. For a list of recommended repositories, please see https://journals.plos.org/ plosone/s/recommended-repositories. You also have the option of uploading thedata as Supporting Information files, but we would recommend depositing data directly to a data repositoryif possible. Please update your Data Availability statement in the submission form accordingly.

A: All the data used in the manuscript, including the biochemical analysis of plasma/blood supernatants and SEM images, have been submitted in the main manuscript and/or the supplementary information. The raw data of supernatant biochemical analysis are shown in Tables 2-5 of the revised version of main manuscript. We would like to draw the attention of the editor/reviewers to the fact that these are raw data and not summary metrics. The summary metrics of the same are shown separately in Figs 2 and 3 of the revised manuscript. Similarly, all the raw post-exposure SEM images (except for the addition of a scale bar; these images are the raw images) of the valves have already been shared as supplementary images in a zip file. Only a subset of SEM images that contained relevant findings that we wanted to highlight in the study have been compiled and presented in the revised version of the main manuscript as Figs 4-7. SEM image measurements of the area occupied by cells/acellular deposits on the valves are summarized in Table 6, while the individual valve measurements are mentioned in the Figure legends of the corresponding valve images (Supplementary images: Figs S1-S25). As we have already shared all the available raw data, we have modified the data availability statement in the PLOS ONE submission portal accordingly, explicitly declaring that we have made all data underlying the findings described fully available, without restriction, and from the time of publication.

A:No recommendation for the citation of specific previously published works has been made by the reviewers.

Reviewer comments

Reviewer #1

This manuscript developed a unique method, the application of voltage, to make the valve antithrombogenic for the valve leaflet. This was a very challenging and interesting topic. However, there are several unclear points in the manuscript that need improvement.

1. Sample photos should be included.

A: Sample photos of the experimental preparation and electrical connectivity of the valve to the pacemaker have been provided as a new Fig 1 [figure caption for the same is at Lines: 135-141, page 6, revised manuscript with tracking]. We have also updated the numbering of other Figures in the main manuscript accordingly.

**Fig 1: The experimental setup for testing the antithrombotic effect and hemocompatibility of electrically activated prosthetic valve . A-Electrical connectivity of the prosthetic valve to the pacemaker with a lead electrode. B-Expanded view of prosthetic valve-lead junction. C-Prosthetic valve with lead was placed in a 50 ml polystyrene test tube for exposure to blood/PRP, while the pacemaker was placed outside the test tube, in a plastic container (shown in D). D-Experimental session with blood/PRP exposure to the valve in an environ shaker and an incubator at 35 ± 2 °C for 30 min.**

2. To deepen the reader's understanding, it would be better to present a scheme of how the valve is intended to be used in the future, including the electrical circuit system.

A: We have created a schematic image to indicate the potential method of deployment in the clinical context in which the valve is proposed to be used, as Fig 8, including pacemaker connectivity, in the discussion section as follows: [figure legend is inserted at Lines: 445-448, page 21, revised manuscript with tracking].

**Fig 8: A schematic representation of the intended clinical deployment of the electrically activated prosthetic valve, including the pacemaker-electrical connectivity. A- Prosthetic mechanical heart valve. B- Valve-lead junction. C-Lead electrode. D-Pacemaker.BioRender was used to create the schematic image.

Lines: 469-471, page 22, revised manuscript with tracking

“ ..A schematic image indicating the potential method of deployment in the clinical context for which the valve is proposed to be used is shown in Fig 8, including pacemaker connectivity…”

We have also shown the valve-pacemaker-electrical circuit pictures in the newly added Figure 1, as mentioned in the answer to comment 1 above.

3. Macroscopic photographs taken after the blood compatibility test should be added to clarify the surface condition.

A: Macroscopic photographs were not captured after the blood compatibility testing, and the specimens were directly sent to the SEM imaging.

4. The SEM photographs show enlarged sections. Please also show the overall picture.

A: All the SEM images, representing the overall picture and enlarged sections of valves, have been submitted in the supplementary information as a zip file. In the main manuscript, we have included a representative panel of 3 SEM images for every valve-voltage condition, corresponding to overall-to-enlarged sections.

5. In the section on limitation, please consider how you think the electrophysiological effects on the heart would be affected if voltage were applied to the valve

A: We have added the following explanation, elaborating on the possible electrophysiological effects on the heart due to electrical activation of the valve, in the limitations section of the discussion: [Lines: 455-469, pages 21-22, revised manuscript with tracking]

“...The in vitro paradigm precludes the assessment of the electrophysiological effects of voltage applied to the valve on the heart. However, the voltages tested in the study (0.25 & 0.5 V) are unlikely to significantly impact the rhythm genesis or propagation in the cardiac muscle for the following reasons: 1. The threshold voltage for myocardial capture for pacing applications is ~1.5 V @ 0.4 ms pulse width, while we have used the pacemaker in a non-pacing mode at 0.25 & 0.5 V for electrical valve activation. The normal vascular endothelium that lines the inside of the heart/blood vessels has a potential difference of ~0.3 V 2. The valve prosthesis will be implanted in the mitral position, which is surrounded by an electrically insulating annulus fibrosus (the physiological fibrous skeleton of the heart). The annulus fibrosis prevents the direct propagation of electrical activity from atria to ventricles in health, and similar behaviour is expected to protect the heart from the electrophysiological spillover effects. 3. The valve prosthesis has a cuff of sewing ring, made up of polyester fabric, which helps suture the valve to the annulus. This fabric material has electrical insulator properties. 4. Peri-prosthetic valve fibrosis, known to occur after valve implantation, further walls off the prosthetic valve electrically from the rest of the heart…”

Reviewer #2

An intriguing and extremely pertinent use of bio-electric engineering is examined in the manuscript "Surface Anticoagulation of Mechanical Heart Valves..." It is a clever idea to reject platelets by imitating the endothelium glycocalyx with a mild electrical current. The reasoning is supported by prior research on surface charge and thrombosis, and the prose is usually understandable. As a reviewer, however, I have serious doubts about how reliable the information is. Formal statistical analysis is not possible due to the study's reliance on a relatively small number of replicates (often one valve per condition). As a result, it is challenging to ascertain whether the reported differences—like the apparent superiority of 0.5 V over 0.25 V—are merely the result of random variation or are biologically significant. To meet the strict requirements of the area, this effort must either strengthen the statistical power or, on the other hand, be far more cautious about the assertions.

1. The abstract mentions comparing control valves to 0.25 V and 0.5 V, but does not specify the sample size(n).

A:We have added the sample sizes in the methods section of the abstract [Lines: 40-42, page 2, revised manuscript with tracking] as follows:

“...Methods: Bileaflet MHVs (n=9) were immersed in human platelet-rich plasma or whole blood, of which and 5 valves were connected to a programmable pulse generator. Control valves (0 V) (n=4) were compared to those activated at 0.25 (n=3), and 0.5 V(n=2)...”

We have also added a new table (Table 1) [Lines: 219-222, page10, revised manuscript with tracking] in the manuscript to clarify the number of valves examined for each voltage × Exposure (blood/PRP) × testing method (Invitro hematology/SEM) condition.

Table 1: The number of valves tested for each voltage, exposure (blood/PRP) and characterization method (invitro/SEM) (n=9). One PRP-exposed valve from each voltage condition (0, 0.25 & 0.5 V) was subjected to SEM imaging.

S.No Method Exposure Control (0 V) 0.25 V 0.5 V

1 Hematologic & coagulation assays Blood 2 2 1

PRP 2 1 1

2 SEM imaging PRP 1 1 1

2. The introduction would benefit from a brief explanation of why the particular voltages of 0.25 V and 0.5 V were selected, even though the background information about the glycocalyx is great. Are these based on safety thresholds for electrochemical corrosion, earlier computer models, or earlier research on vascular grafts (Sawyer et al.)?

A: We have added the following content in the last paragraph of the introduction section to justify our choice of voltages at 0.25 & 0.5 V [Lines 103-116, page 5, revised manuscript with tracking]

“...For a proof of concept, we explore the antithrombotic effect at two specific voltages i.e., 0.25 and 0.5 V. The choice of these voltages is guided by the following reasons: 1. The normal vascular endothelium in humans is antithrombotic at a potential of 0.3 V due to negatively charged residues in glycocalyx. The choice of 0.25 & 0.5 V is motivated in part by the proximity of these values to the physiological endothelial potential in health. 2. The present work is based on the assembly of a prosthetic mechanical heart valve and the commercially available pacemakers (with leads). The lowest voltage of the commonly used pacemakers is 0.25 V (St. Jude pacemaker, now Abbott) or 0.5 V (Medtronic). 3. Higher voltage at the valves is likely to increase the risk of myocardial capture as the threshold pacing voltage is approached, potentially affecting rhythm genesis or propagation. 4. Higher voltage also increases the rate of battery discharge, shortening the duration of antithrombotic efficacy when used in a clinical context, requiring frequent pulse generator replacement…”

3. This is the critical weak point of the manuscript. Section 2.6 states, "formal statistical comparisons were not performed." In a comparative efficacy study, this is a significant deviation from standard practice.

A: We concede that the lack of formal statistical comparison is a limitation of the study. However, unlike the clinical trials, it is not customary or feasible for statistical power to be explicitly factored in for the proof-of-concept, investigator-initiated, pre-clinical in vitro or in vivo studies in the domain of prosthetic valve research due to the limited number of feasible replicates. The present study should be seen as an exploratory study to nucleate further investigation into this promising and potentially revolutionary approach to safe long-term anticoagulation for prosthetic heart valve implantation. Moreover, the Committee for the Purpose of Control and Supervision of Experiments on Animals (CPCSEA), under the Ministry of Fisheries, Animal Husbandry and Dairying, Govt. of India, has approved an invivo study on a swine model (No. V-11011(13)/6/2024-CPCSEA-DADF, dated: 08-07-204), based on the the invitro results reported in this work. The invivo study will attempt to replicate, validate, and extend the findings reported in this paper, including the long-term effects.

4. The valves were exposed to blood/PRP for only 30 minutes. While platelet adhesion occurs early, meaningful thrombus formation and fibrin network development on MHVs typically occur over longer periods.

A: We agree with the reviewer's comments that thrombus formation and fibrin network development on MHVs typically occur over a longer period. However, this is an in vitro study, which was conducted as per the ISO 10993:4 (2017) International Standard for Blood Material Interaction. In an in-vitro study, blood/PRP exposure cannot be prolonged, as it can cause diffuse blood cell activation, precluding any meaningful analysis and interpretation of the test results. According to the cited standard, the time of exposure for blood or PRP can be upto a maximum of 30 min-1hour for an in vitro study. Furthermore, it is also recommended to complete the analysis within 4 hours of blood collection. Also, long-term effects like thrombus formation/fibrin network deposition are consequences of the initial cellular and protein absorption response. Thus, platelet adhesion and activation, which are immediate responses, predict thrombus formation in the long term. As discussed in the answer to comment #3 above, an in vivo study has been approved and is currently in progress to investigate the long-term effects of blood/PRP exposure to electrically activated prosthetic heart valves.

5. Hemolysis was found to be less than 0.1%, which is a favorable safety signal. But was the donor's hematocrit taken into account while calculating the free hemoglobin?

A: Haemolysis was also tested as per the ISO 10993:4. As per the standard, the hemoglobin of the donor should be more than 12.5 gm/dL. The method used for free Hemoglobin was spectrophotometry. The hematocrit (Hct) was not considered for the calculation of the % hemolysis reported in the data of the manuscript (Equation 2 belo

---

## [Decision Letter · Decision Letter 1]

8 Apr 2026

Surface Anticoagulation of Mechanical Heart Valves using Electrically Induced Biomimetic Glycocalyx: an In-vitro study to assess Hemocompatibility and Optimal Voltage

PONE-D-25-57739R1

Dear Dr. Sajja,

We’re pleased to inform you that your manuscript has been judged scientifically suitable for publication and will be formally accepted for publication once it meets all outstanding technical requirements.

Kind regards,

Jeevithan Elango, PhD

Academic Editor

PLOS One

Additional Editor Comments (optional):

Thanks for the revised version.

Reviewers' comments:

Reviewer's Responses to Questions

**Comments to the Author**

1. If the authors have adequately addressed your comments raised in a previous round of review and you feel that this manuscript is now acceptable for publication, you may indicate that here to bypass the “Comments to the Author” section, enter your conflict of interest statement in the “Confidential to Editor” section, and submit your "Accept" recommendation.

Reviewer #1: All comments have been addressed

Reviewer #2: All comments have been addressed

2. Is the manuscript technically sound, and do the data support the conclusions?

Reviewer #1: Yes

Reviewer #2: Yes

3. Has the statistical analysis been performed appropriately and rigorously? 

Reviewer #1: Yes

Reviewer #2: Yes

4. Have the authors made all data underlying the findings in their manuscript fully available?

Reviewer #1: Yes

Reviewer #2: Yes

5. Is the manuscript presented in an intelligible fashion and written in standard English?

Reviewer #1: Yes

Reviewer #2: Yes

6. Review Comments to the Author

Reviewer #1: (No Response)

Reviewer #2: (No Response)

7. PLOS authors have the option to publish the peer review history of their article (what does this mean?). If published, this will include your full peer review and any attached files.

Reviewer #1: No

Reviewer #2: No

---

## [Editor Report · Acceptance letter]

PONE-D-25-57739R1

PLOS One

Dear Dr. Sajja,

I'm pleased to inform you that your manuscript has been deemed suitable for publication in PLOS One. Congratulations! Your manuscript is now being handed over to our production team.

Kind regards,

on behalf of

Dr. Jeevithan Elango

Academic Editor

PLOS One